# A Cluster Analysis of Risk Factors for Cancer across EU Countries: Health Policy Recommendations for Prevention

**DOI:** 10.3390/ijerph18158142

**Published:** 2021-07-31

**Authors:** Dawid Majcherek, Marzenna Anna Weresa, Christina Ciecierski

**Affiliations:** 1Collegium of World Economy, SGH Warsaw School of Economics, 02-554 Warsaw, Poland; 2World Economy Research Institute, Collegium of World Economy, SGH Warsaw School of Economics, 02-554 Warsaw, Poland; mweres@sgh.waw.pl; 3Department of Economics, Northeastern Illinois University, Chicago, IL 60625, USA; c-ciecierski@neiu.edu

**Keywords:** cancer policy, cancer risk factors, socioeconomic status, cancer prevention, the European Union, clustering

## Abstract

Cancer burden in the European Union (EU) is increasing and has stimulated the European Commission (EC) to develop strategies for cancer control. A common “one size fits all” prevention policy may not be effective in reducing cancer morbidity and mortality. The goal of this paper is to show that EU member states are not homogenous in terms of their exposure to risk factors for cancer (i.e., lifestyle, socio-economic status (SES), air pollution, and vaccination). Data from a variety of sources including Eurostat, the UNESCO Institute for Statistics, the European Health Interview Survey, Eurobarometer, and the European Environment Agency were merged across years 2013–2015 and used to develop a cluster analysis. This work identified four patterns of cancer prevention behaviors in the EU thus making it possible to group EU members states into four distinct country clusters including: sports-engaged countries, tobacco and pollutant exposed nations, unhealthy lifestyle countries, and a stimulant-enjoying cluster of countries. This paper finds that there is a need for closer collaboration among EU countries belonging to the same cluster in order to share best practices regarding health policy measures that might improve cancer control interventions locally and across the EU.

## 1. Introduction

Health economics and its relation to cancer prevention continues to be a major public health priority for Europe and remains one of the five research missions in the forthcoming Horizon Europe, the European Union’s Framework Programme for Research and Innovation for years 2021–2027 [1]. Cancer has been a growing challenge for the EU due to the large and growing burden of cancer incidence. Europe, which represents around 10% of the world’s population, is home to a quarter of the world’s cancer cases [2]. According to data from Eurostat, the number of deaths due to cancer in the EU-27 exceeded 1,2 million in 2016, an increase of 13% since 2001 [3]. Despite an appreciable decline in the overall average mortality rate in the EU-27, from 267.9 per 100,000 inhabitants in 2011 to 257.06 in 2016 [3], the value of this indicator differs significantly across EU states, ranging from 211.84 deaths per 100,000 inhabitants in Cyprus to 342.14 in Hungary. Moreover, there are diverse trends in the standardized death rates caused by cancer in the EU. For example, while there has been no significant recent change in cancer deaths in Portugal and Lithuania, the death toll from cancer in Cyprus, Romania, Bulgaria, and Greece has been rising since 2011. In further comparison, Luxembourg and Belgium noted double-digit declines between 2011 and 2017 (Figure 1). Figure 1 compares standardized death rates due to cancer (horizontal axis) with the dynamic of the death rate (vertical axis) among EU states over the period 2001–2017. Indeed, EU countries can be divided into four groups in terms of cancer deaths rate and their changes over time:Group 1: death rates lower than the EU-27 average and are decreasing. These include 11 EU Member states: Luxembourg, Malta, Spain, Austria, Belgium, Italy, Germany, Finland, Sweden, Portugal, and France;Group 2: includes Bulgaria, Greece and Cyprus which present with death rates that are lower than the EU average, but which are increasing;Group 3: consists of 11 countries including: Ireland, Denmark, the Czech Republic, the Netherlands, Croatia, Hungary, Slovakia, Slovenia, Estonia, Lithuania, Latvia, and Poland—which present death rates that are higher than the EU average, but are decreasing.Group 4: captures only one country (Romania) where the death rate due to cancer is much higher than the EU-27 average and this mean value continues to rise.

Given this evidence, the question arises: why do disparities in cancer survival rates occur across the EU? Many studies of risk factors for cancer reveal that new cancer cases, and in turn, cancer-related deaths could be avoided by reducing exposure to lifestyle and environmental risk factors [4,5,6,7,8]. This further begs the question: can preventative cancer policy be applied to curb cancer mortality in the EU? What should EU cancer control policies entail? Disparities in death rates due to cancer and their related variability across EU countries indicate that national cancer prevention programs should be customized and remain relevant to the patterns in cancer risk factors the prevail within a given EU state. Therefore, the main aim of this paper is to identify patterns in risk factors for cancer across EU member states and to propose recommendations for cancer prevention policy as they relate to those patterns.

This paper is structured into five sections which include: an introduction, a review of previous studies regarding cancer determinants to support the appropriate selection of critical modifiable risk factors relevant to cancer prevention, a data and methods section followed by the presentation of research results as well as a discussion section that compares our findings with other studies on this topic. The final section concludes the results and delineates policy implications.

## 2. Risk Factors for Cancer—A Review of Previous Studies

In order to implement appropriate and adequate cancer policy prevention measures, precise knowledge about the causes of cancer is essential. Cancer occurs as a consequence of exposure to carcinogenic agents, of which some are related to genotype while others are shaped by the environment in which people live. Although we have little ability to shape genetic determinants of cancer, studies show that factors related to human behavior (including but not limited to tobacco consumption, alcohol use, diet, and physical activity) are significant to the etiology of cancer [8]. As stated earlier in this paper, the number of new cancer cases and the probability of cancer survival differ substantially across the EU. As such, there is a critical need to better understand cancer risk factors, with particular focus awarded to those risk factors that are modifiable as these largely depend on changing the surrounding environment and/or human behavior. Approximately 30–45% of cancer cases could be prevented through the promotion of healthier lifestyles [8,9]. The Global Burden of Diseases Study provides evidence that shows how lifestyle risk factors attribute to disability-adjusted life-years (DALYs) [10].

The monograph series for the Identification of Carcinogenic Hazards to Humans and Handbooks of Cancer Prevention published by the International Agency for Research on Cancer (IARC) of the World Health Organization classifies all agents that are carcinogenic to humans by organ site and based on scientific evidence, presents a list of interventions that can have cancer-preventive effects [11,12,13,14,15,16,17,18,19,20,21,22,23,24,25,26,27]. The causes of cancer can be grouped into the following broad categories: tobacco products, infectious agents, alcohol consumption, exposure to sunlight and ultraviolet radiation, subjection to ionizing radiation and radiofrequency electromagnetic fields, diet and nutrition, insufficient physical activity, obesity, dietary carcinogens, contamination of air, water, soil, and food, some pharmaceutical drugs (e.g., diethylstilbestrol and phenacetin), and submission to occupational carcinogens [28]. According to the IARC monographs, interventions with sufficient evidence of creating cancer-preventive effects include smoking cessation, the absence of excess body fat, regular physical activity, and reductions in alcohol consumption. Anti-cancer screening programs are also associated with reductions in the incidence of fatal cancers [11,12,13,14,15,16,17,18,19,20,21,22,23,24,25,26,27,28,29]. In addition, the literature also shows that inoculations play a critical role in cancer prevention. Several studies reveal that therapeutic cancer vaccines based on viral antigens (i.e., vaccinations for the human papilloma virus (HPV) and hepatitis B) have been effective in fighting precancerous lesions [30,31,32,33]. Thus, increasing the population’s participation in vaccination programs is recommended for inclusion in national cancer prevention policies [29].

Most strategies for cancer prevention in the EU not only call for the promotion of healthy lifestyle but also advocate for environmental protection and adjustments to health care services through the implementation of both integrated and palliative care [8]. As such, environmental risk factors should also be addressed. The increased risk of environmental pollution on cancer incidence has been widely reported in the literature [34,35,36]. For example, the European Environment Agency (EEA) stresses the negative consequences of air pollution proving that it creates the single largest environmental health risk in Europe. Indeed, it has been estimated that every year PM_2.5_ (Particulate Matter) causes over 300,000 premature deaths in the EU-28 [37]. The IARC classifies air pollution, and in particular PM, as carcinogenic [38].

A review of the literature suggests that while cancer prevention policy must focus on the behavioral and environmental risk factors discussed above, cancer is also strongly associated with social and economic status [39]. Socioeconomic inequities result from a variety of factors, such as occupation, income level, [40,41], educational attainment, societal structure [35], economic structure, and policy [42]. Exposure to various risk factors for cancer, such as tobacco, alcohol, unhealthy diet, and physical activity are highest for low socio-economic populations. The profile of cancer types also differs between high- and low-income countries. Moreover, patient access to innovative medicine differs between wealthier and poorer nations [9]. However, with rising levels in socio-economic development, exposure to various risk factors also change. On the one hand, higher income translates into higher standards of living as well as better sanitation and hygienic conditions resulting in decreases of infection-related cancer. On the other hand, however, a rapid rise in the economic growth rate is often accompanied by deterioration of the natural environment, and increased exposure to cancer risk factors, such as pollution, may in turn, yield higher rates of other types of cancer incidence [28].

To date, no studies have taken a less homogenous approach to examining the modifying influence of lifestyle, environmental and socioeconomic factors to decrease potentially preventable cancer cases and related deaths among specific groups from across the EU member states. This study uses a cluster analysis approach to take into account uneven exposure to risk factors for cancer as well as the heterogeneity of contextual factors among individual member states in order to categorize EU member states into homogenous clusters. In turn, such clusters of the population might better respond to targeted cancer prevention policies rather than a single, common framework for the EU.

Taking into account this context, the following research questions can be asked in relation to appropriate and effective policy aimed at cancer prevention in the European countries:To what extent are EU member states homogenous in terms of exposure to various primary cancer risk factors?How should policy handle lifestyle and environmental factors that are potentially modifiable in order to decrease potentially preventable cancer cases and related deaths?What are the elements of cancer prevention policies that constitute a common and complete framework for the EU? Which policies should vary across countries, taking into account uneven exposure to risk factors for cancer and the heterogeneity of contextual factors in individual member states?How to bridge insufficient awareness of best practices in cancer prevention actions and foster their implementation in the different EU Member States?

## 3. Data and Methods

### 3.1. Data

The following country-level data sources were used for this analysis:-Cancer mortality
∘Cancer deaths taken from the Global Burden of Disease Study 2017 [43] (variable name: cancer ratio, data from 2015)-Socio – Economic status (SES):
∘GDP per capita in EUR from Eurostat [44] (variable name: GDP per capita, data from 2015)∘Completed years of education from The UNESCO Institute for Statistics (UIS) [45] (variable name: yearsofedu, data from 2015)∘Domestic general government health expenditure as a percentage of Gross Domestic Product (GDP) (%) derived from the World Health Organization (WHO) [46] (variable name: healthcare (HC) spending, data from 2015)-Lifestyle:
∘Alcohol consumption in liters from the WHO [47] (variable name: alcohol, data from 2014)∘% of people who smoke currently from the European Health Interview Survey (EHIS) [48] (variable name: smoke, data from 2014)∘% of population which eats fruits or vegetables more than 5 times weekly from European Health Interview Survey (EHIS) [48] (variable name: diet, data from 2014)∘% of population with a Body Mass Index (BMI) >= 30 from EHIS data [48] (variable name: obese, data from 2014)-Sports Activity:
∘% of population with membership to a sports club; from Eurobarometer [49] (variable name: sports club membership (SC), data from 2013)∘% of population which exercises or engages in sports at least one time per week; from Eurobarometer [49] (variable name: sports activity (SA), data from 2013)-Air Quality:
∘Particulate Matter (PM) statistics from the European Environment Agency (EEA) [50] mainly PM_10_, which are inhalable particles, with diameters that are generally 10 micrometers and smaller (variable name: PM_10_days—number of days when PM_10_ exceeds 50 µg/m^3^)-Vaccination:
∘Human papillomavirus (HPV) vaccination coverage across EU [51] (variable name: HPV segment, data from 2006 to 2017) aggregated to several levels:
No data≤30% (very low)31–50%51–70%≥71% (high)

Based on the availability of data, a total of 27 European Union (EU) countries were analyzed and included: Austria, Belgium, Bulgaria, Cyprus, Czech Republic, Germany, Denmark, Estonia, Greece, Spain, Finland, France, Croatia, Hungary, Ireland, Italy, Lithuania, Luxembourg, Latvia, Malta, Netherlands, Poland, Portugal, Romania, Sweden, Slovenia, and Slovakia.

### 3.2. Methods

The cluster analysis among EU countries based on SES, lifestyle, sports activity, HPV vaccination, and air quality was conducted using the K-means Clustering Method with implementation of Euclidean distance and the standardization of variables [52]. Standardization of variables was performed in order to limit potentially large effect on results, which might yield complications when variables with large variances are used. The aim of this analysis was to identify groups of countries with as many similarities as possible with regards to their respective risk factors for cancer and cancer deaths (within clusters) yet different from one another in other aspects. According to prior studies, there is no single decision criteria available as to how to select an adequate number of clusters [53,54,55]. The selection of clusters is a multi-decision problem and additional algorithms must be developed to automatically resolve this issue. We empirically determined that the 4-cluster solution yielded the best match, because with this split, all clusters were disjoint sets and empirical interpretation was reasonable. The cluster analysis was conducted using R [56] and the CRAN fact extra package [57].

## 4. Results

The merged dataset containing information regarding cancer deaths, SES, lifestyle, sports activity, HPV vaccination, and air quality made it possible to distinguish four clusters from across the EU states. Appendix A in the annex presents the values of all indicators for the individual country states. The following clusters were identified:Cluster I (8 countries): Belgium, Germany, Denmark, Finland, Ireland, Luxembourg, the Netherlands, and SwedenCluster II (9 countries): Bulgaria, Cyprus, Greece, Croatia, Hungary, Latvia, Poland, Romania, and SlovakiaCluster III (4 countries): Spain, Italy, Malta, PortugalCluster IV (6 countries): Austria, the Czech Republic, Estonia, France, Lithuania, and Slovenia

Cluster results are presented in Figure 2 and Table 1.

Cluster I captures countries with the lowest rates of cancer deaths per 100,000 inhabitants (i.e., from 192 cases/100,000 in Ireland to 291 cases/100,000 in Germany). The most notable difference with respect to other clusters is SES. While this cluster is characterized by an average number of completed years of education of approximately 13, the most notable difference is in GDP per capita, which measures an average of roughly 49,395 EUR. This is twice as high as in Clusters III and IV and thrice that of Cluster II. In addition, public health care (HC) expenditures amount to around 7.3% of GDP which is the highest among all clusters. HPV vaccination coverage amounts to approximately 51–70% and represents the second highest cluster in terms of coverage. Moreover, the air pollution indicator, PM_10_days, is the lowest among clusters. Lifestyle variables vary significantly across all clusters. For example, medium-ranked levels of alcohol consumption (liters per capita) across all clusters range from 7.16 L in Sweden to as high as 11.99 L in Germany. Smoking rates are lowest (one percent) in Sweden and as high as 16 percent in the Netherlands. Medium-ranked levels of fruit and vegetable intake exceed 5 times per week (i.e., from 2% in Germany, up to 30% in Sweden) and the number of obesity deaths is lowest (i.e., from 13% in the Netherlands to 18% in Ireland). This cluster includes countries with the highest sports activity (SA), from 47% in Belgium to as high as 70% in Sweden, and the highest presence of sports club (SC) membership (i.e., from a low of 12% in Finland to a high of 27% in the Netherlands).

Cluster II is characterized by the highest of all cancer deaths rates (i.e., from a low of 223 deaths/100,000 in Cyprus to 346 deaths/100,000 in Hungary). Countries belonging to Cluster II are also those with the highest relative levels of air pollution. For example, the PM_10_ days measure ranges from 14.53 in Romania to 63.63 in Bulgaria. Moreover, these EU States stand out from other clusters because of their lowest levels of SES (including the lowest values of GDP per capita, second lowest number of years of education completed, lowest HC expenditures, and lowest HPV vaccination coverage). In addition, the SA measure is low (i.e., ranging from a low of 11% sports activity in Bulgaria to only 38% in Hungary) as is the measure for SC membership (i.e., 1% in Romania and a high of only 10% in Croatia). Cluster II also includes countries with highly varying lifestyles. While smoking rates are comparable to that of Cluster IV, alcohol consumption and dietary behaviors differ significantly. The average rate of alcohol consumption is closely comparable to those in Cluster I (i.e., from 9.55 liters in Cyprus to 11.30 L in Bulgaria) while smoking rates are alarmingly high and span from 20% in Romania to 31% in Bulgaria. This cluster also captures relatively low prevalence rate of vegetable or fruit intake (i.e., only 1% of the population in Romania and 29% of the population in Greece eat fruits or vegetables five or more times per week). Obesity rates are relatively average but vary significantly within the cluster (i.e., from 9.1% in Romania to almost 21% in Hungary and Latvia).

Cluster III is characterized by the second to lowest rates in cancer deaths (252 deaths per 100,000), a relatively poor healthy lifestyle and limited participation in sports activity. Countries belonging to Cluster III are states with medium-ranked GDP per capita (i.e., an average of only 22,362 EUR per annum), medium levels of spending on HC and the lowest number of years of education (i.e., from a low of 9.11 years in Portugal to a high of 11.25 years in Malta). HPV vaccination coverage is appreciably high as the ratio exceeds 71% in all countries except Italy, and the number of days when PM_10_ values exceed EU limits is fairly average. Cluster III presents with the lowest levels of alcohol consumption (roughly 8.42 L) and the second to lowest rates of smoking (an average of approximately 12% incidence). However, the rate of vegetable or fruit intake is also very low. In this cluster only 6% of the population in Italy and 18% of the people of Portugal consume fruits or vegetables 5 times per week of more. The incidence of obesity in this cluster is an average of 17.1% which is the highest from across clusters (ranging from 10% in Italy up to 25% in Malta) while SA percentages are among the lowest with average of 31% engaging is sports activity (and ranging from a low of 19% in Malta to a high of 46% in Spain) and an average of only 6% of the population holding membership to sports clubs.

In countries belonging to Cluster IV, the rate of cancer deaths per 100,000 is the second highest among clusters and SES and lifestyle vary widely. In this cluster, the average number of completed years of education is relatively high (i.e., from a low of 11.4 in France to 13.9 years in Estonia). GDP per capita is almost half of that in Cluster I and similar to Cluster III. The percentage of GDP dedicated to HC spending is second highest, but HPV vaccination coverage is less than 30%. The air pollution measure, PM_10_ days, ranks at medium levels while lifestyle measures differ significantly from other clusters. That is, Cluster IV countries differentiate themselves appreciably from the others through their high alcohol consumption (i.e., 11.4 L in Austria to 16.64 L in Estonia) and comparatively high smoking rates (from 14% in Slovenia to 26% in Estonia). On the one hand, this cluster is characterized by a high percentage of people who eat fruits or vegetables more than 5 times per week (from 11% in Lithuania to 31% in the Czech Republic), but also, a high prevalence of obesity (from 14% in Austria to almost 20% in Estonia). Cluster IV includes countries with high average ranges in sports activity participation (i.e., the SA measure ranges from 36% in Czech Republic to 51% in Slovenia) while sports club membership varies from only 8% in Lithuania to 16% in France.

## 5. Discussion

We identified four patterns of cancer prevention behaviors in the EU, which made it possible to group EU members states into four distinct clusters: (1) sports-engaged countries, (2) tobacco and pollutant exposed countries, (3) states characterized by unhealthy lifestyles, and (4) a stimulant-enjoying cluster of countries. In general, our results confirm the existing evidence regarding risk factors for cancer, and in particular, those primary factors do include tobacco and alcohol, diet, obesity, insufficient physical activity, exposure to air pollution, and low levels of vaccination. Our results also reveal striking variations between EU member states, particularly when considering in the importance of factors that can reduce cancer risk as well as cancer death rates within EU states. These differences indicate that many lives could be saved if countries move to applying locally relevant policies to control cancer.

Having established a clear need for policy differentiation, the next step included an analysis of differences between EU countries. Expectedly, Cluster I (characterized by highest SES, lowest air pollution levels and highest rates of sports activity) also reports the lowest levels of cancer deaths (248 deaths per 100,000 inhabitants). We find that prevention policy among countries grouped together in Cluster I (sports-engaged countries) should address lower alcohol consumption and healthy diet. In light of these results, it is not surprising that Cluster II reports highest rates of cancer mortality. Lowest SES, highest air pollution levels, low HPV vaccination, highest smoking rates, and lowest sports activity in Cluster II are associated with an almost 10% higher cancer mortality rate per 100,000 inhabitants relative to Cluster I. Given this, it is critical to establish a broad policy awareness campaign in order to modify lifestyles and limit air pollution among Cluster II states (i.e., a tobacco and pollutant exposed cluster). Interestingly, despite low sports activity rates and unhealthily diet, Cluster III presents with the second lowest cancer death rate (252 deaths per 100,000 inhabitants). However, Cluster III is also characterized by high levels of HPV vaccinations, lowest alcohol consumption, and low smoking rates. This combination of factors and stable SES are associated with the second lowest cancer mortality rates in the EU. In contrast, high alcohol consumption, high smoking rates, and very low HPV vaccination rates are the main reasons for why Cluster IV states present with the second highest cancer death rates (261 deaths per 100,000 inhabitants). Interestingly, Cluster IV reports highest fruit and vegetable consumption while its obesity rates are among the highest in the EU. Given this, alcohol intake may be a risk factor for obesity in Cluster IV. Perhaps the most important conclusion from this cluster analysis is that the EU is not homogenous in terms of exposure to various primary risk factors for cancer and so cancer prevention policies should be more fitted to cluster and country-specific needs.

Our research indicates that cancer control policies should be tailored to the specific country profile that results from exposure to various cancer risks, successful implementation of control measures such as vaccinations as well as contextual factors reflected in the socio-economic status of the population. Our comprehensive analysis of cancer prevention policy in the EU countries showed that nearly all EU countries, except for Bulgaria, Croatia, and Slovakia, have some form of cancer control policies or strategies in place, but that these are at different stages of their implementation. There is a large variety of policy approaches in the EU and a shortfall exists in the sharing of best practices and their implementation. Therefore, a common EU policy framework is needed coupled with guidelines in order to assess and compare the effectiveness of cancer prevention policies across the EU [8]. This finding is supported by the analysis of eleven different areas of health policies (including cancer) in the EU member states, which reveal a variety of approaches, but also show gaps in the research regarding the assessment of policy impacts and outcome indictors [58]. There have been important differences between countries located in the same European region, and these diverging trends in health performance in Europe can be, to a large extent, attributed to the successes and/or failures of health policy [58]. Our analysis goes beyond these conclusions, showing the potential to share best practices promote learning within country clusters by watching what other countries with similar cancer risk profile did to control cancer, what worked, and what did not. This paper focuses on cancer prevention policy thus we are not discussing broadly correlation between different cancer risk factors. However, Appendix A section provide an overview of the association between cancer mortality and various risk factors for cancer among all EU countries. Indeed, cancer mortality in the EU is positively related with smoking and air pollution while negatively correlated with sports activity, diet, and SES. As Mackenbach and McKee conclude, “considerable health gains could be achieved if all countries would follow best practice in health policy”. We do not fully agree with this conclusion and subsequently argue that when and where cancer prevention policy is concerned, one size does not fit all. Sharing best practices and their successful implementation makes sense only when countries have similar cancer risk patterns.

## 6. Contribution and Limitations

The contribution of this paper is three-fold. First, our analysis goes beyond the traditional factors that impact the probability of cancer incidence as analyzed in the literature to date (i.e., tobacco and alcohol use, diet, sports activity) by including air pollution and HPV vaccinations, and also taking into account the SES of the population. Second, this paper proves that the EU Member states are not homogenous in terms of risk factors for cancer and identifies the patterns of these factors allowing to group EU member states into four different clusters, which have different profiles of exposure to cancer risk. Third, our findings have important implications for policy and shed new light on modifiable risk factors for cancer by underlining those which are of key importance for identified clusters within the EU. This knowledge seems to be necessary for cancer prevention strategies, which can be more effective if introduced with sensitivity and understanding of the characteristics of population behavior within a particular country. The set of anti-cancer policies relevant for four county clusters is discussed in the next section and is illustrated in Table 2.

The analysis conducted in this paper is not without limitations, which are related to the conceptual assumptions and data used. As far as paper conceptualization is concerned, our analysis has taken into account all types of cancer. We are aware that some cancer risk factors may be more relevant for some cancer types and less relevant for others. For instance, alcohol consumption has been linked to cancer of the oral cavity, pharynx, colorectum, and liver, but there is limited evidence for pancreatic cancer along with other evidence that suggests a lack of carcinogenicity for kidney cancer [59,60]. This analysis would be more precise if cancer cases and death rates were broken down by cancer type. However, such detailed data combined with lifestyle data stratified by gender are not always available. Furthermore, some risk factors (e.g., tobacco and alcohol) act synergistically causing increased risk [61], which was not taken into account in our analysis. Our study did not include data regarding cancer screening programs, but this could only be done in a more detailed analysis prepared for a specific cancer type.

Another limitation of this research is related to variables that we employed for identifying country clusters. We used cancer mortality to assess cancer burden following some other studies on this topic [6,8,35]. Cancer incidence is another metric that could have been used, however, with both incidence and mortality rates highly correlated (Pearson’s correlation coefficient = 0.99—see Appendix A), we chose cancer mortality. Employing the other (cancer incidence) would have likely not significantly altered the results (see Appendix A with more data about cancer incidence).

Analysis of cancer risk factors is also dependent on the completeness of the data sets. There was missing data in relation to HPV vaccination, which were not under data imputation. Thus limited our analysis and indicated the need for further coordination of data collection at the EU level. Moreover, Eurobarometer data concerning sports activity derives from 2013 while the majority of the remaining data was collected in 2015.

Further research on this topic may address these limitations, given that the relevant data is available. There is also a need to develop our analysis by adding a regional perspective as differences exist not only across the EU, but also within individual countries ass proved for the case of Poland by Majcherek et al. [62]).

## 7. Conclusions and Policy Implications

Many elements of a cancer control programs are in place in EU countries including attempts to build intelligence through interdisciplinary research based on cancer in the EU and under the umbrella of the mission-oriented research entitled “Conquering Cancer” introduced to the EU Framework Program entitled “Horizon Europe” for the years 2021–2027 [1]. However, apart from integrating all efforts to fight cancer, there is also a need to create a comprehensive national system tailored to specific characteristics of any given, individual EU countries. Our analysis indicates that there are some similarities in cancer risk factors in EU member states, which allow to group them into four clusters presented in Figure 3: sports-engaged cluster (1), tobacco and pollutant exposed (2), a cluster dominated by unhealthy lifestyle (3), and a stimulant-enjoying cluster (4). As a result, cancer prevention policy should be different for each cluster being adjusted to its specific pattern of risk factors. Policies for cancer prevention should have various foci in each of these country clusters and address the most important cancer risk factors. An overview of policy target areas is presented in Table 2.

Furthermore, our analysis shows that establishing closer collaborations among EU countries that belong to the same clusters and aim to share experiences and best practices concerning the efficiency of various policy measures might be beneficial for cancer control in the EU, particularly for achieving common synergies and overall health gains.

## Figures and Tables

**Figure 1 ijerph-18-08142-f001:**
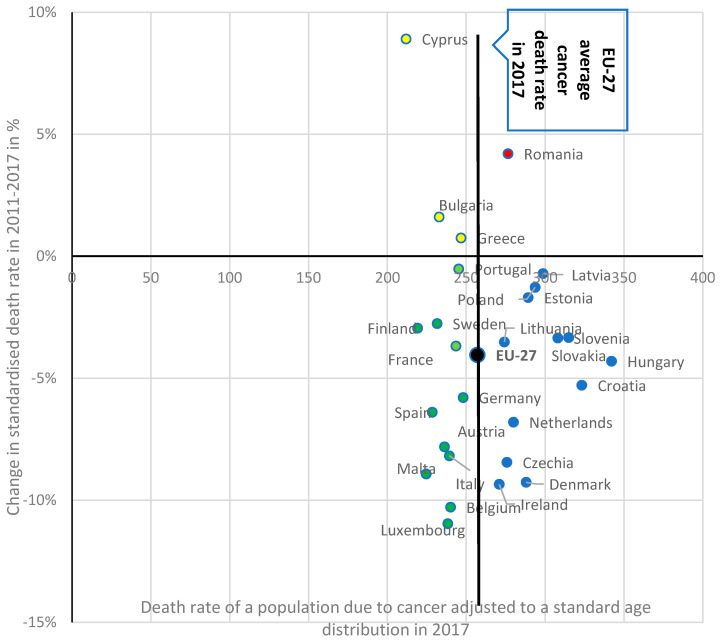
Death rates of population due to cancer adjusted to a standard age distribution per 100,000 inhabitants in the European Union Member states in 2017 (or latest available year) and their changes between 2011 and 2017 (in %). Source: Authors’ elaboration based on Eurostat data extracted on 2 February 2021 (last update of data: 21 October 2020 [3]).

**Figure 2 ijerph-18-08142-f002:**
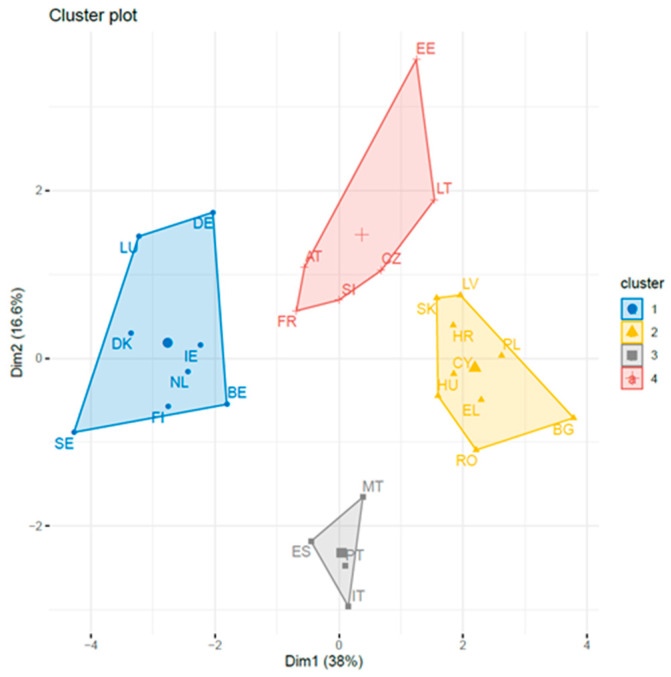
Cluster plot for 11 indicators of cancer deaths and 27 countries.

**Figure 3 ijerph-18-08142-f003:**
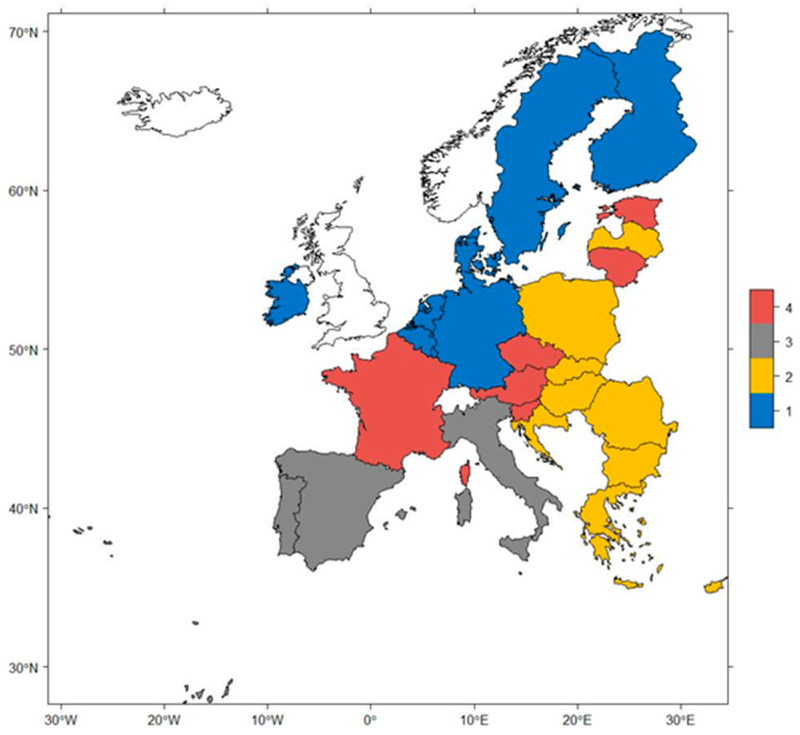
Clustering results presented on the map of Europe. Source: Author’s elaboration.

**Table 1 ijerph-18-08142-t001:** Cluster means for 11 indications and cancer deaths.

**Cluster**	**CancerRatio [per 100k]**	**GDP/Capita [EUR]**	**HC Spending [%]**	**Years of Edu**	**PM_10_days**	**HPV Segment**
1	248.05	49,395.00	7.30	12.64	8.13	4.25
2	274.92	12,456.67	4.34	11.97	34.62	2.00
3	252.19	22,362.50	6.18	10.13	17.96	4.75
4	261.21	22,731.67	6.25	12.67	15.62	2.00
**Cluster**	**Alcohol** **[liters]**	**Smoke [%]**	**Diet [%]**	**Obese [%]**	**Sports Club [%]**	**Sports Activity [%]**
1	9.77	8.13	15.63	15.24	0.21	0.58
2	10.52	25.11	12.22	16.26	0.05	0.29
3	8.42	12.25	12.00	17.00	0.06	0.31
4	13.11	22.00	21.17	17.10	0.12	0.42

Note: CancerRatio is cancer mortality per 100,000 inhabitants; GDP is Gross Domestic Product; HC spending is Public Healthcare expenditures as a % of GDP; PM_10_ represents Particulate Matter with a diameter between 2.5 μm and 10 μm; HPV segment is Human Papilloma Virus vaccination coverage segment.

**Table 2 ijerph-18-08142-t002:** Differences among four clusters of EU countries in cancer prevention policy focuses.

Cluster	Name	Cancer-Risk Factors—Focus Areas
1 (blue)	Sports-engaged cluster	alcohol consumption; fruit and vegetable intake
2 (yellow)	Tobacco and pollutant exposed cluster	SES; air pollution; smoking rates; fruit and vegetable intake; sports activity; HPV vaccination
3 (grey)	Unhealthy lifestyle cluster	fruit and vegetable intake; obesity; sports activity; air pollution
4 (red)	Stimulant-enjoying cluster	alcohol consumption; smoking rates; obesity; HPV vaccination

Source: Author’s elaboration.

## Data Availability

The study was conducted based mainly on publicly available data described in section Data and methods with references. Moreover, D.M. and M.A.W. received confidential data from Eurostat from European Health Interview Survey (EHIS) under Research Project Proposal RPP 117/2020-EHIS.

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
