# Peer review of "A Cluster Analysis of Risk Factors for Cancer across EU Countries: Health Policy Recommendations for Prevention"

_ijerph, 2021, doi:10.3390/ijerph18158142_

Round 1
Reviewer 1 Report
Majcherek et al. present a cluster analysis of cancer mortality and the established modifiable cancer risk factors across EU counties. The premise and objectives behind the research is sound and the methods are reasonable with a few caveats. The analysis and results are well presented and some, but not all, limitations are discussed.
The authors explore the association between cancer and the established modifiable life-style cancer risk factors across EU countries. The findings are of potential use in guiding policies aiming at cancer prevention. The data analysed were collected from several public sources, that are well documented by the authors. The cluster analysis was employed in the data analysis. The authors identified four specific clusters with distinct pattern of associations between cancer mortality rates and life-style modifiable cancer risk factors. The main conclusion drawn from the analysis was that cancer prevention strategies across EU countries may be more beneficial if they are country/region specific.
Comments/recommendations to the authors:
1) The authors discuss some, but not all, limitations in their analysis. While the main goal relates to informing policy makers on cancer prevention, it is a concern that cancer mortality, rather than cancer incidence is the primary metric used. Especially, since the type of cancer is not available. Cancer survival prognosis differs dramatically depending on the type of cancer. In addition, prevalence of different cancers may differ between the countries. The rational or justification should be made for the use of cancer mortality beyond the fact of data unavailability.
2) HPV vaccination is anticipated to decrease mortality of (primarily) cervical cancer once adopted and in place in a country/region over the long term (decades). This reviewer finds it difficult to believe that the current EU uptake would have any impact on total cancer mortality in any region at this point in time. If this is not something the authors agree upon, then the they should present numbers (deaths from cervical cancer etc.) from EU or other world regions that argue their case, and justify why HPV vaccination is likely to be a useful risk factor for inclusion in their analysis. Apart from that, it would be useful for the authors to rerun the cluster analysis omitting HPV vaccination to determine the impact of HPV segment has on defining the four cluster regions.
3) The lack of information on cancer screening per country is a major shortfall of the study.
4) The authors provide a detailed description of the four identified country clusters and the different patterns of association between the life-style risk factors and cancer mortality among the clusters. However, there are no quantitative estimates for the strength of the association between the different life-style factors and cancer (or cancer mortality in this case). This seems an important point and should be addressed by the authors.
5) Words such as “lead to” and “leading to” (lines 336, 343 and 345) overstate the claims and “associated with” is more appropriate.
6) In some areas, like page 3, lines 126-130, the narrative seems to go off track, i.e. not specifically related to cancer. The reference to COVID on page 3, lines 113-114 is totally unwarranted.
7) The ranges of references on page 3, lines 98 and 107-108 perhaps need to be merged. There is an extra unwarranted bracket on page 4, line 147.
I see no major issues. There is a misspelling on Page 3, line 92 and the missing spaces in the title of the Supplementary Table S1.
Author Response
Dear Editor, Dear Reviewers,
In response to the evaluation of our manuscript entitled “A Cluster Analysis of Risk Factors for Cancer across EU Countries: Health Policy Recommendations for Prevention”, we would like to thank the Editors and Reviewers for all of their valuable comments and suggestions which have undoubtedly added value and significantly improved the quality of the submitted article.
In the manuscript file, changes have been implemented using the MS Word ‘track changes’ option so that edits are easily identifiable. Replies to Reviewer comments are provided below. Our replies directly follow each reviewer comment and begin with the word “Author Response:”.
We thank you for the opportunity to resubmit this article and appreciate your time and effort in considering this manuscript. We hope that in its corrected format, this article meets editorial requirements, proves valuable to the existing literature, presents clearly to the audience, and is suitable for publication in the special issue of the International Journal of Environmental Research and Public Health entitled Socio-Economic Factors of Cancer in Health Economics section.
Responses to the comments received from Reviewer #1:
1) The authors discuss some, but not all, limitations in their analysis. While the main goal relates to informing policy makers on cancer prevention, it is a concern that cancer mortality, rather than cancer incidence is the primary metric used. Especially, since the type of cancer is not available. Cancer survival prognosis differs dramatically depending on the type of cancer. In addition, prevalence of different cancers may differ between the countries. The rational or justification should be made for the use of cancer mortality beyond the fact of data unavailability.
Author Response: Thank you for your comment. We agree that cancer survival prognosis differs across various types of cancer. This is explained in the Contributions and Limitations Section (lines 448-459 in track changes) of the manuscript. In its current version, the manuscript also includes a new paragraph dedicated to a discussion of study limitations. Among the topics addressed is our use of cancer mortality rather than cancer incidence as a metric of cancer burden (lines 462-468 in track changes). Indeed, the literature reflects the use of both metrics. In this paper, we chose to use one of the two (cancer mortality) as using both in our cluster analysis would not be methodologically appropriate. We chose to include cancer mortality as it is a more acute measure than incidence. Indeed, the best option would be to run separate cluster analyses using cancer mortality and next, cancer incidence and later, compare the results. However, manuscript restriction will not allow us to extend our analysis with this additional measure. Moreover, we completed a Pearson’s correlation test for mortality and incidence in the EU countries. The results of this test have been added to the Supplementary materials of this paper. Kindly refer to Figure B for an illustration of both cancer incidence and cancer mortality rates. The resulting Pearson’s correlation coefficient is relatively high (0.99). As such, we find that either metric would yield similar cluster identified.
2) HPV vaccination is anticipated to decrease mortality of (primarily) cervical cancer once adopted and in place in a country/region over the long term (decades). This reviewer finds it difficult to believe that the current EU uptake would have any impact on total cancer mortality in any region at this point in time. If this is not something the authors agree upon, then the they should present numbers (deaths from cervical cancer etc.) from EU or other world regions that argue their case, and justify why HPV vaccination is likely to be a useful risk factor for inclusion in their analysis. Apart from that, it would be useful for the authors to rerun the cluster analysis omitting HPV vaccination to determine the impact of HPV segment has on defining the four cluster regions.
Author Response: We agree that HPV vaccination carries impacts in the long run. In fact, the majority of our risk factors for cancer (i.e. diet, smoking, drinking, air quality, etc.) have long run impacts on cancer mortality.
According to the WHO, cervical cancer is the fourth most frequent cancer among women and accounts for 7.5% of all female cancer deaths. (https://www.who.int/news-room/fact-sheets/detail/human-papillomavirus-(hpv)-and-cervical-cancer). Scientific evidence also shows that HPV can cause other types of cancer (e.g. mouth, throat, anus) (Chaturvedi AK. Beyond Cervical Cancer: Burden of Other HPV-Related Cancers Among Men and Women. Journal of Adolescent Health 2010;46(4):S20-S26. doi: 10.1016/j.jadohealth.2010.01.016). For example, according to the United States (US) National Cancer Institute (NCI), 70% of oropharyngeal cancers in the US are caused by HPV and the number of new cases is rising. Moreover, oropharyngeal cancers are the most common HPV-related cancer in the US (https://www.cancer.gov/about-cancer/causes-prevention/risk/infectious-agents/hpv-and-cancer#cancers-caused). In the Europe, about 2.5% of all cancer incidence is attributable to HPV (A Four Step Plan for Eliminating HPV Cancers in Europe, European Cancer Organization, 2020, p. 8).
We chose to include HPV vaccination in the study analysis in order to draw attention to the importance of this primary preventative measure, which is included as such in The Global Strategy Towards Eliminating Cervical Cancer adopted by the WHO in 2020 and with EU guidelines (https://www.ecdc.europa.eu/en/all-topics-z/human-papillomavirus/scientific-advice-human-papillomavirus).
A recent study by Brisson et al. (2020) entitled “Impact of HPV vaccination and cervical screening on cervical cancer elimination: a comparative modelling analysis in 78 low-income and lower-middle-income countries” and published in The Lancet, 395(10224), 575–590. https://doi.org/10.1016/s0140-6736(20)30068-4) shows that high HPV vaccination coverage can lead to the elimination of cervical cancer in low- and lower-middle-income countries by the end of this century. Moreover, the results of the 10-year HPV program in Australia has proven a large impact of the HPV vaccine in reducing several HPV-related diseases in Australia (See: Patel et al. (2018). The impact of 10 years of human papillomavirus (HPV) vaccination in Australia: what additional disease burden will a nonavalent vaccine prevent? Eurosurveillance, 23(41). https://doi.org/10.2807/1560-7917.es.2018.23.41.1700737 ). Because cervical cancer is responsible for 2.5% of all cancer incidence, elimination of HPV-related cancers may have positive impact on the overall measures of cancer incidence and cancer mortality.
We thank the Reviewer for the suggestion to rerun the cluster analysis with omitting the variable for HPV vaccination. In follow-up to this suggestion, we plan to include this type of analysis in our future research agenda.
3) The lack of information on cancer screening per country is a major shortfall of the study.
Author Response: We are very grateful for the reviewer’s attention to this important problem. We are aware that such data exists for EU countries (https://ec.europa.eu/health/sites/default/files/major_chronic_diseases/docs/2017_cancerscreening_2ndreportimplementation_en.pdf). However, it is difficult to assess quality of this data and its current status among screening programs. We were not able to add cancer screening to our cluster analysis because of the difficulty of merging our existing database with a single measure that would capture a variety of screening programmes conducted in different years for a variety of cancer types in just one overall screening result for all cancer cases. To mitigate this shortfall in our study, we added a sentence in the limitations section explaining this issue (lines 459-460 in the track changes version)
4) The authors provide a detailed description of the four identified country clusters and the different patterns of association between the life-style risk factors and cancer mortality among the clusters. However, there are no quantitative estimates for the strength of the association between the different life-style factors and cancer (or cancer mortality in this case). This seems an important point and should be addressed by the authors.
Author Response: Thank you for this important comment. It is difficult to analyse a cause-and-effect relationship between particular factors, primarily of because of the low sample size when running for instance, linear regression models. However, through Figure A in the Supplementary materials, we present a correlation matrix. An additional table, Table S2 in the Supplementary Tables has been inserted to present the strength of the correlation between cancer mortality and risk factors for cancer. Figure A shows that cancer mortality is positively associated with smoking and air pollution while being negatively associated with GDP per capita. The only significant relationship, however, is between GDP per capita and the Cancer Ratio. As presented in our earlier study (Majcherek, D.; Weresa, M.A.; Ciecierski, C. Understanding Regional Risk Factors for Cancer: A Cluster Analysis of Lifestyle, Environment and Socio-Economic Status in Poland. Sustainability 2020, 12, 9080, doi:10.3390/su12219080) any interpretation of direction and strength of relationship should be made with caution.
5) Words such as “lead to” and “leading to” (lines 336, 343 and 345) overstate the claims and “associated with” is more appropriate.
Author Response: Wording in the text has been adjusted accordingly.
6) In some areas, like page 3, lines 126-130, the narrative seems to go off track, i.e. not specifically related to cancer. The reference to COVID on page 3, lines 113-114 is totally unwarranted.
Author Response: Thank you for your comment. Indeed, the Literature Review section of the paper has been considerably shortened to include only the most pertinent of earlier studies. The reference to the COVID-19 vaccination on page 3 has been removed.
7) The ranges of references on page 3, lines 98 and 107-108 perhaps need to be merged. There is an extra unwarranted bracket on page 4, line 147.
Author Response: The references have been adjusted accordingly throughout the text.
8) I see no major issues. There is a misspelling on Page 3, line 92 and the missing spaces in the title of the Supplementary Table S1.
Author Response: Wording, including spelling errors, have been adjusted accordingly throughout the text of the manuscript.

Reviewer 2 Report
The manuscript “A Cluster Analysis of Risk Factors for Cancer across EU Countries: Health Policy Recommendations for Prevention” described four patterns of cancer prevention behaviors in the EU.The goal of this paper is to prove that EU member states are not homogenous in terms of their exposure to risk factors for cancer,to share best practices regarding health policy measures that might improve cancer control in the EU. I believed that the authors carefully carried out their study and the results could support their conclusions. The manuscript may be considered for publication in International Journal of Environmental Research and Public Health after minor revision, so long as the following minor issues were carefully considered and responded.
1. It is recommended to modify the graphic abstract. It is not novel enough.
2. It is suggested to add more charts.
3. Please carefully check the format of the article and check for spelling mistakes and grammatical issues.
Author Response
Dear Editor, Dear Reviewers,
In response to the evaluation of our manuscript entitled “A Cluster Analysis of Risk Factors for Cancer across EU Countries: Health Policy Recommendations for Prevention”, we would like to thank the Editors and Reviewers for all of their valuable comments and suggestions which have undoubtedly added value and significantly improved the quality of the submitted article.
In the manuscript file, changes have been implemented using the MS Word ‘track changes’ option so that edits are easily identifiable. Replies to Reviewer comments are provided below. Our replies directly follow each reviewer comment and begin with the word “Author Response:”.
We thank you for the opportunity to resubmit this article and appreciate your time and effort in considering this manuscript. We hope that in its corrected format, this article meets editorial requirements, proves valuable to the existing literature, presents clearly to the audience, and is suitable for publication in the special issue of the International Journal of Environmental Research and Public Health entitled Socio-Economic Factors of Cancer in Health Economics section.
Responses to the comments received from Reviewer #2:
- It is recommended to modify the graphic abstract. It is not novel enough.
Author Response: We have decided to include the most important and pertinent of information in the written abstract, upon which we rely more heavily than on the graphic representation, which is less significant. Nevertheless, we have also expanded the manuscript’s supplementary materials section to include additional charts that present more specific data to motivate and more adequately capture the background and context of this research study.
- It is suggested to add more charts.
Author Response: Thank you for this suggestion. We have added two new charts and one additional table to the supplementary materials section of the manuscript. First, the new Table S2 presents the Pearson correlation coefficients with p-values for all possible pairs of variables used in our analysis. Second, the New Figure B shows depicts cancer deaths and cancer incidence per country for 2017 along with information about the Pearson correlation coefficient between cancer mortality and cancer morbidity at the level of 0.99. Finally, new Figure C presents cancer incidence calculated per 100 000 inhabitants.
- Please carefully check the format of the article and check for spelling mistakes and grammatical issues.
Author Response: The wording and spelling of the text has been checked and edited through the manuscript.

Reviewer 3 Report
A Cluster Analysis of Risk Factors for Cancer across EU Countries: Health Policy Recommendations for Prevention by Majchereket al.
The overall research question is interesting.
Few suggestions:
Figure one can be improved with different color codes for different groups.
It is not well explained how 4 groups are formed – in the introduction, Introduction seems to be extra-long and includes unnecessary information as well. Can trim down a bit.
Table 1 – explain abbreviations
The whole manuscript can be trimmed down with necessary information only.
Author Response
A Cluster Analysis of Risk Factors for Cancer across EU Countries: Health Policy Recommendations for Prevention
Dear Editor, Dear Reviewers,
In response to the evaluation of our manuscript entitled “A Cluster Analysis of Risk Factors for Cancer across EU Countries: Health Policy Recommendations for Prevention”, we would like to thank the Editors and Reviewers for all of their valuable comments and suggestions which have undoubtedly added value and significantly improved the quality of the submitted article.
In the manuscript file, changes have been implemented using the MS Word ‘track changes’ option so that edits are easily identifiable. Replies to Reviewer comments are provided below. Our replies directly follow each reviewer comment and begin with the word “Author Response:”.
We thank you for the opportunity to resubmit this article and appreciate your time and effort in considering this manuscript. We hope that in its corrected format, this article meets editorial requirements, proves valuable to the existing literature, presents clearly to the audience, and is suitable for publication in the special issue of the International Journal of Environmental Research and Public Health entitled Socio-Economic Factors of Cancer in Health Economics section.
Responses to the comments received from Reviewer #3:
- Figure one can be improved with different color codes for different groups.
Author Response: Thank you for your suggestion. We have inserted an improved version of Figure 1 so as to use different colours for the four groups (purple, yellow, green and blue).
- It is not well explained how 4 groups are formed – in the introduction, Introduction seems to be extra-long and includes unnecessary information as well. Can trim down a bit
Author Response: Thank you for your comment. An explanation of how the 4 groups/clusters were constructed has been clarified in the text. Please see lines 42-47 and the corresponding Figure has been greatly improved through the addition of new colour coding.
- Table 1 – explain abbreviations
Author Response: We appreciate your comment. The full name of each abbreviated variable name has been added directly under Table 1.
- The whole manuscript can be trimmed down with necessary information only.
Author Response: Thank you for your suggestion. Indeed, the Literature Review section of the paper has been considerably shortened to include only the most pertinent of earlier studies.
